# Cervical Cell Image Classification-Based Knowledge Distillation

**DOI:** 10.3390/biomimetics7040195

**Published:** 2022-11-10

**Authors:** Wenjian Gao, Chuanyun Xu, Gang Li, Yang Zhang, Nanlan Bai, Mengwei Li

**Affiliations:** 1School of Artificial Intelligence, Chongqing University of Technology, Chongqing 400054, China; 2College of Computer and Information Science, Chongqing Normal University, Chongqing 401331, China

**Keywords:** deep learning, cervical cells, image classification, transfer learning, knowledge distillation, self-distillation, context information, ensemble

## Abstract

Current deep-learning-based cervical cell classification methods suffer from parameter redundancy and poor model generalization performance, which creates challenges for the intelligent classification of cervical cytology smear images. In this paper, we establish a method for such classification that combines transfer learning and knowledge distillation. This new method not only transfers common features between different source domain data, but also realizes model-to-model knowledge transfer using the unnormalized probability output between models as knowledge. A multi-exit classification network is then introduced as the student network, where a global context module is embedded in each exit branch. A self-distillation method is then proposed to fuse contextual information; deep classifiers in the student network guide shallow classifiers to learn, and multiple classifier outputs are fused using an average integration strategy to form a classifier with strong generalization performance. The experimental results show that the developed method achieves good results using the SIPaKMeD dataset. The accuracy, sensitivity, specificity, and F-measure of the five classifications are 98.52%, 98.53%, 98.68%, 98.59%, respectively. The effectiveness of the method is further verified on a natural image dataset.

## 1. Introduction

Cervical cancer is a common malignant tumor in women and a serious threat to their life and health [1]. However, with the popularization of national screening for “two cancers”, the increasing number of cervical cytology smears screened has placed tremendous pressure on hospitals. Not only is manual film reading labor-intensive, but interpretations are also subjective and this can easily lead to misdiagnosis. Therefore, the study of cervical cytology image classification algorithms has important academic value and social significance.

With the development of artificial intelligence, many researchers have turned to machine learning to improve computer-aided diagnosis ability [2,3,4,5]. In the cervical cell classification task, this usually requires manually designed features, such as morphology, color, and texture, which are then input into the classifier after feature extraction. In [6], morphological features, such as circumference and the area of cervical cancer cells, were selected and support vector machine (SVM) was used to classify the cells according to specific features. However, because the algorithm is very dependent on manually designed features, too many or too few features were selected for it to perform effectively.

In general, the more obvious features of abnormal cervical cells are enlarged nuclei and significant changes to the nucleoplasmic ratio. Therefore, a two-stage segmentation approach followed by classification was used in [7] to classify cells. The images were first enhanced using Gaussian filtering and histogram equalization. The cell nuclei were then segmented using adaptive threshold segmentation based on the weighted Otsu method. Finally, the extracted nuclei area and gray value features were classified using the K-means clustering algorithm. This segmentation-then-classification approach is more dependent on segmentation accuracy, and the above algorithms are not suitable for solving multi-class classification problems because they all use a single classifier for classification of cervical cancer cells. Some studies have implemented multi-classification tasks by combining multiple classifiers, such as a combination of SVM and K-nearest neighbor algorithms, multiple SVM cascades and AdaBoost, to improve accuracy in the multi-classification of cervical cells. The above studies have all made outstanding contributions to cervical cell classification, but the shortcoming remains that the two-stage segmentation approach followed by classification conducts predication according to the accurate segmentation of cells, thereby not only ignoring the global semantic information, but also increasing the training cost. Deep-learning-based classification methods can effectively overcome these problems, but existing studies using such methods have commonly used large-scale deep neural network models, resulting in model parameter redundancy. Cervical cell datasets are mostly privatized and, because few are publicly available, there is also a problem of their limited number and the difficulty for a single network to learn all feature views, resulting in low generalization of the model.

Therefore, to overcome the above problems, the contributions of this paper are summarized as follows:We develop a transfer method that combines knowledge distillation and transfer learning. Based on the generic features of migration data, the category information learned by the teacher network is distilled into the student network to further improve the performance of the student network.The developed method is proposed for application in cervical cell classification. A multi-export classification network with different depths was constructed to capture different levels of features, a self-distillation loss function was used to enhance the classification performance of the shallow network and an ensemble strategy was used to fuse the prediction results of multiple exits.A global context module is introduced in each exit branch to enable the classifier to capture different fine-grained features. This supplements the contextual information while differentiating the sub-models and ensures the effectiveness of the integration.Compared with high-latency large-scale networks, self-distillation is integrated on traditional knowledge distillation in this paper. Not only model compression is achieved, but performance improvement is also considered.

## 2. Related Work

### 2.1. Cervical Cell Classification

In recent years, convolutional neural networks (CNNs) have been successfully applied to the multi-classification of cervical cells and have demonstrated powerful performance. Such networks have the advantage of being end-to-end classifiers with no manual feature design required and the network can automatically learn easily distinguishable cell features from the input cervical cancer cell images. For example, [8] overcame the shortcomings of traditional cervical cell classification algorithms by introducing a CNN based on depth features. In [9], a new cervical cell dataset containing five classes of single-cell data was proposed that used SVM, multilayer perceptron (MLP) and a CNN to classify single cells without pre-segmentation.

Deep neural networks are gradually becoming the mainstream approach for cervical cell classification tasks. In [10], 94.89% accuracy in classification was achieved using the ResNet152 network without the need for segmentation and manual feature extraction. In [11], parallel convolutional layers were added to a CNN to enhance its feature extraction capability, which, in turn, improved cervical cell classification. Many studies have started to focus on improving neural network performance. For example, to improve the classification performance of cervical cell images, in [12], a squeeze-and-excitation block and a spatial attention block were introduced to fuse the channel and spatial features. In [13], ResNet50 was selected as the backbone network to which a feature pyramid pooling layer and a long short-term memory module (LSTM) were added and its effectiveness was experimentally demonstrated.

However, large-scale deep neural network models are commonly used in existing approaches, resulting in redundancy of model parameters. Moreover, the generalization performance of the models needs to be improved. In this paper, multiple sub-models with different depths are used to classify the samples and a multi-exit self-distillation method is constructed to enhance the classification performance of the network.

### 2.2. Transfer Learning and Knowledge Distillation

In supervised learning tasks, the number of samples with annotations often directly affects the final results of a deep model. To overcome the problem of insufficient data, many studies have introduced transfer learning into the medical imaging domain [14,15,16]. The model weights that are learned from natural image data, or other data in the medical field, are usually migrated to the target model and the model weight parameters are then fine-tuned using task-specific datasets to achieve feature sharing, thus improving the classification performance of the network model. In the initial stages of the COVID-19 pandemic, there was a paucity of lung CT images of positive samples, so [17] used a CNN that had been pre-trained on a natural image dataset (ImageNet), which was then used in COVID-19 detection, achieving 98% accuracy. A multi-source data transfer method to improve model performance was proposed in [14]. The model was first pre-trained on source data for heart and breast images. The knowledge from the pre-trained model was then migrated for use in the task of diagnosing Alzheimer’s disease. The experimental results demonstrated that the method effectively improved diagnostic accuracy and significantly reduced model training time. In [16], transfer learning was combined with an attention mechanism as the basis for the proposal of a transfer learning method. By migrating the knowledge from three sources of data and using the attention mechanism to improve the feature extraction ability, the final model achieved an accuracy of 96.63%. All the above studies demonstrate the effectiveness of transfer learning.

The intention of knowledge distillation (KD), an important method for knowledge transfer and model compression, is to allow simple models to learn effective information from complex models and to obtain an approximate performance for complex models [18,19,20,21,22]. This structure is often visualized as a teacher–student network, an idea first proposed in [18]. A poorer student network improves the performance of its network by learning from the teacher network, provided that an experienced teacher network is available. Subsequently, [19] proposed a method for the student network to learn the output of the teacher network using soft labels, which they called “knowledge distillation”. The conventional knowledge distillation method learned only from the output of the teacher network, which led to the loss of intermediate layer knowledge. Therefore, subsequently, many researchers tried to use the knowledge of intermediate, relational, and structural features to exploit the maximum potential of the teacher network [20,21,22,23,24,25]. Self-distillation is a new approach that was developed from knowledge distillation. Unlike traditional knowledge distillation architectures, the teacher-student architecture of self-distillation uses the same model [26] or a network framework without teachers [27]. In [28], neural networks were used to study knowledge distillation from a new perspective; instead of compressing the model, the student network was optimized based on a teacher network with equivalent parameter settings. By constructing a multi-exit student network, Xu et al. [29] proposed a distillation structure combining knowledge distillation and self-distillation to improve the performance of the student network. Applying knowledge distillation to cervical cell classification to improve model generalization is a worthwhile approach.

In addition to the above, an alternative approach has involved fusing the knowledge of data and model features through distillation combined with transfer learning. Knowledge distillation and transfer learning share similar ideas regarding transfer, but they play complementary roles. Traditional transfer learning migrates the knowledge of data features while ignoring the knowledge of model features; moreover, according to the work of Z et al. [30], individual networks cannot usually learn all view features due to their limited “capacity”. Therefore, based on transfer learning, we introduced a knowledge distillation algorithm, i.e., we constructed a teacher–student network to transfer category information learned in the teacher to the student network and improved its performance. In this paper, we combine the two methods to classify cervical cells, which involves deep integration of data and model feature knowledge aimed at reducing the training cost and enhancing the model performance.

## 3. Proposed Method

### 3.1. Transfer Learning and Knowledge Distillation Fusion

Figure 1 shows the overall framework of the cervical cell classification method based on the fusion of knowledge distillation and transfer learning. It is divided into three aspects: domain data, teacher–student model, and knowledge transfer. Among these, the data are sourced from two different datasets: the natural image and cervical cell datasets. The models were two independent and heterogeneous teacher–student networks. The knowledge transfer part includes data transfer and model transfer. The training process can be seen from the figure that natural image data and cervical cell image data enter the network in two stages. The first stage is to train the initialized model using natural images. The second stage is to use the cervical cell image data to fine-tune the pre-trained network to achieve the purpose of model weight reuse. Then comes the distillation stage, which freezes the teacher network and does not participate in training, and only uses the soft tags output by the teacher network and the soft tags of the student network to calculate the loss function, so as to achieve the purpose of supervising the student network by the teacher network, thereby improving the student network. performance.

In detail, the student network is used as the target model for cervical cell classification and is involved in training and optimization throughout. The teacher network only provides soft label information and the performance of the final method depends on the classification accuracy of the student network. The loss function contains the following two components: classification loss with cross-entropy loss, which uses real labels for supervised learning, and distillation loss with Kullback–Leibler divergence, which is used to measure the distance between the unnormalized probability distributions of the teacher network and the student network output using the category probabilities of the teacher network output for guided learning. During training, the two are jointly optimized for the student network.
(1)$${\mathrm{Loss}}_{CE}=\frac{1}{n}\sum_{i}^{n}L_{i}=-\frac{1}{n}\sum_{i}^{n}\sum_{c=1}^{m}y_{ic}\log\left(p_{ic}\right),$$
where *n* denotes the number of samples; *i* is the category index, with m categories; $y_{ic}$ and $p_{ic}$ represent the true probability that observation sample *i* belongs to category *c* and the predicted probability of the model, respectively. When the class of the observation sample *i* is *c*, $y_{ic}$ equals 1, and is 0 otherwise. The class probability of the network prediction is obtained from the fully connected layer output $z^{c}$ according to the SoftMax function:(2)$$L_{KL}\left[X^{s}∥X^{t}\right]=\sum_{i}^{n}\mathrm{softmax}\left(\frac{x_{i}^{s}}{T}\right)\log\frac{\mathrm{softmax}\left(\frac{x_{i}^{s}}{T}\right)}{\mathrm{softmax}\left(\frac{x_{i}^{t}}{T}\right)}$$

In the above equation, *T* is the hyperparameter of knowledge distillation, which represents the distillation temperature and controls the softness of the logits output: the higher the temperature, the smaller the relative value between the category probabilities.

### 3.2. Multi-Export Student Network with Contextualized Information

Lightweight student networks have difficulty learning all the features used for correct classification due to their limited capacity. This results in poor generalization, which affects the classification performance in the test set. Previous studies have ignored the fact that knowledge distillation can enhance the model through self-learning. In this paper, we propose a multi-export network incorporating contextual information and self-distillation as a student network to enhance the model’s ability to classify cervical cell images. By introducing classifier branches at different stages of the student network, different fine-grained classification features are learned. The final results of multiple sub-model predictions are then obtained through an integration strategy to improve the model generalization. Meanwhile, to enhance the feature extraction performance of each branch and ensure that the sub-classifiers focus more attention on the different local area features, the global context module and the self-distillation algorithm are integrated in each branch network to improve the performance of shallow classifiers and achieve improved performance overall.

The multi-export network structure is a design method that inserts multiple classifiers at different depths. Classifier exports at different depths have different feature-learning capabilities for the correct classification of simple samples at shallow layers, thus enabling samples to end predictions earlier and saving computational costs.

Each classifier exit is a sub-model with different convolutional and fully connected layers as classifiers, wherein deep classifiers demonstrate better performance than shallow classifiers. Therefore, a multi-export-based self-distillation model was proposed with its deep classifier exports as the teacher network to provide supervised information for guided learning to improve shallow classifier performance. The so-called self-distillation refers to doing it alone in a network, using the exits of different levels as sub-models, and optimizing from deep to shallow through the distillation loss function.

The improved multi-export network structure in this paper is based on the dynamic network (SCAN) proposed in [31] and teacher-student collaborative knowledge distillation (TSKD) in [26]. As shown in Figure 2, it is divided into three parts: a backbone network, global context module, and a shallow classifier. ResNet18 was chosen for the backbone network and classifier exits were inserted after different stages. Without significantly increasing the computational effort, each sub-export is introduced into the GC block, which consists of three functions: modeling the global attention mechanism for the input features, performing feature transformation, and fusing the input features by the dot product. The features output from each sub-export pass through the bottleneck layer to unify the feature map scales and then pass through the fully connected layer to output the probability for each category. In the training phase, the final output of the backbone network guides shallow classifier learning via self-distillation such that the high-level semantic knowledge learned by the deep classifier can be transferred to the shallow classifier to improve the generalization performance of each exit model. Furthermore, the deep classifier achieves integration of low-level and high-level semantics via the global context building block. The final classification results were obtained by integrating multiple exit prediction probabilities during testing.

### 3.3. Loss Function

Given N data samples of class C, for input samples $x\in {\left\{x_{i}\right\}}_{i=1}^{N}$, $z^{k}$ represents the output of the fully connected layer for the category index *k*, and the output of the teacher model for the category k probability is expressed as
(3)$$t^{k}=\frac{\exp\left(z^{k}/\tau \right)}{\sum_{k}^{C}\exp\left(z^{k}/\tau \right)}.$$

The loss function has two components: the loss of regular knowledge distillation, which contains the KL dispersion between the teacher and the student, and the loss of cross-entropy between the student output and the true label. The second part refers to the self-distillation loss, which treats the deepest classifier (Exi-n) in the multilevel classifier as a second teacher, using rich unnormalized probabilities and features as supervised information for the shallow level.
(4)$${\mathrm{Loss}}_{KD}^{m}=\tau^{2}\cdot L_{\mathrm{KL}}\left(S_{m},t\right)+L_{\mathrm{CE}}\left(S_{m},y\right)+L_{\mathrm{CE}}\left(S_{n},y\right)$$
(5)$${\mathrm{LosS}}_{SD}^{m}=\tau^{2}\cdot L_{\mathrm{KL}}\left(S_{m},S_{n}\right)+\lambda \cdot {∥\mu_{m}\left(F_{m}\right)-F_{n}∥}^{2}$$

In the above equation, $S_{m}$ and $S_{n}$ represent the soft labels of the *m*-th shallowest exit classifier and the backbone network classifier outputs in the student network after temperature, respectively. Similarly, $F_{m}$ and $F_{n}$ represent the feature outputs before the fully connected layer in the *m*-th and the deepest exit branches of the student network, respectively. $u_{m}$ (·), representing the bottleneck layer structure, was added to each exit network to ensure that the scales of the two remain consistent. The mean squared loss function was used here to minimize the difference between the feature distribution of the shallower network and the deepest convolution. λ is defined as the weight of the L2 loss. Thus, the overall loss of the student network can be expressed as
(6)$$\mathrm{Loss}=\sum_{m=1}^{n}\left({\mathrm{Loss}}_{KD}^{m}+{\mathrm{Loss}}_{SD}^{m}\right).$$

For testing, an average integration algorithm was used for the student network to incorporate the different classification performances of multiple outlets. This is different from multi-teacher network integration and multi-student collaborative distillation, which integrate the multi-level outputs of the student network itself without introducing additional models, which effectively reduces model complexity. In the following equation, $S_{m}$ represents the *m*-th classification exit output and weight and *f* represents the final output of the model.
(7)$$f=\frac{1}{n}\sum_{m=1}^{n}S_{m}$$

### 3.4. Ensemble Strategy

In this paper, because the classifiers were constructed in different stages of the student network with different perceptual fields, they learned different fine-grained features; these tiny local features and global features are the keys to discriminating cervical cells. Therefore, the introduction of multi-export integration averaged the prediction probabilities of multiple exits to obtain the model prediction results, which reduced the variance in the prediction probabilities of multi-export classifiers, decreased the model error rate, and formed a student network that integrated local and global features. The multiple exit classifiers from the student network constitute different learners. A strong classifier was formed by averaging and summing to output the final prediction probabilities.

### 3.5. Global Context Module

The effectiveness of integration is reliant on the differences between multiple classifiers to achieve differentiation across the models and to capture the context-dependent information of each exit. A global context module was introduced into the trunk branch to extract features of interest to each sub-model based on local and global information [32]. The shallow branch is more concerned with local low-level semantic information, while the deep branch is concerned with global high-level semantic information. Therefore, each sub-model differs in its ability to fit different data and the differentiated models effectively exploit the integration potential. The structure of the global context block was divided into three modules: context building to obtain global context-related features, feature transformation to capture the dependencies between channels, and fusion to fuse global context features into features at all locations.

## 4. Experimentation

To verify the effectiveness and advancement of the proposed method, the experimental approach comprised the following three elements: comparison experiments with existing representative algorithms, ablation experiments to separately verify the effectiveness of the numerous proposed strategies, and further validation on a public natural image dataset.

### 4.1. Dataset and Implementation Details

The experiments were performed using the SIPaKMeD [9] cervical cell dataset, which contains 4049 single cell images from 966 pap smears containing clusters of manually cropped cells. These were classified into five categories based on morphology and appearance, as shown in Table 1 and Figure 3.

The publicly available cervical cell dataset contains information about free single cells that could be directly used for the classification task. Data preprocessing begins with dividing into training and validation sets and then randomly dividing these for the training, validation, and test sets in a ratio of 3:1:1. Due to the limited training samples, data augmentation was the most direct way to increase sample diversity and improve data complexity, while overcoming the problem of overfitting. Particularly for the cervical cell dataset, there was a serious lack of available data, so it was difficult to verify the robustness of the algorithm. Therefore, a series of data-enhancement methods, such as random crop scaling, horizontal flip, mirror flip, and rotation, were selected to expand the dataset samples before training the model.

The model optimization method used stochastic gradient descent (SGD). The total number of training epochs was 180, the experimental hyperparameters including the number of the batchsize were 128, and the initial value of the dynamic learning rate was set to 0.1, which was multiplied by 0.1 in the 100th and 150th epochs. The weight decay and momentum factors were 1 × 10${}^{-4}$ and 0.9, respectively, the temperature τ was 3, and the loss weight λ was 0.03, which was empirically determined.

### 4.2. Comparison Experiments

To verify advancement through use of the method proposed in this paper, it was compared with mainstream methods in the field. The experiments used ResNeSt50 as the teacher network and ResNet18 as the student backbone network [33]. Students were trained until convergence and independent predictions were then performed. Table 2 shows the classification results using the SIPaKMeD dataset. It can be seen from the prediction results that the proposed method showed improvement compared to conventional CNNs, graph neural networks [34] and traditional classification algorithms, such as multilayer perceptron and support vector machines [9]. It also achieved 98.52% five classification accuracy based solely on an improved 18-layer residual network, which represented a 0.72% improvement over the optimal outcomes described in [35]. Furthermore, the number of parameters of the improved multi-export student network was 12.33 M, which is smaller than the teacher network and the large-scale networks used in the literature. The original idea of knowledge distillation was for model compression, rather than focusing on improving performance. While self-distillation is a special case, in this paper, we combine knowledge distillation and self-distillation combined methods. The specific approach is to change the student network to a multi-exit network on the traditional distillation architecture. While the teacher network provides supervision information, it optimizes itself through self-distillation. It not only improves the performance of the model, but also achieves a certain compression rate.

### 4.3. Ablation Experiments

The knowledge distillation method used for cervical cell classification in this paper incorporates t-learning (TL), knowledge distillation (KD), and multiple exit self-distillation (SD) methods. Ablation experiments were performed to verify the effectiveness of each component. As shown in Table 3, the first stage of the classification method incorporating transfer learning and knowledge distillation showed a 2.35% improvement over the baseline method. In addition, both transfer learning and knowledge distillation methods improved classification performance, but to different degrees. This indicates that using both generic feature knowledge among data and category information knowledge among models has a positive effect on cervical cell classification. In the second stage, the student network was expanded into a multi-export network that incorporates contextual information. Its performance improved by a further 0.86% because of self-distillation and integration algorithms, which demonstrates that the improved multi-export student network has better classification performance than the original network.

To further illustrate the effectiveness of the improved student network without adding the migratory learning method, but still using ResNet18 as the main backbone network, experiments were carried out to compare the benchmark method with the original multi-export network regarding performance. The classification accuracy for each sub-classifier and integration is shown in Table 4. The output results for each exit and after integration are shown using a confusion matrix in Figure 4. Where the benchmark method did not use any policy, the constructed multiple classifiers were all supervised using only true labels and the prediction results of the final integrated multiple classifiers. The dynamic network SCAN is a scalable neural network framework that uses different levels of classifiers for different classification tasks [31]. Multiscale dense neural networks (MSDs), on the other hand, insert classifiers at different depths of the convolutional network and use dense connections for their optimization [36]. In this paper, alternatively, a multi-export network incorporating contextual information is proposed based on the interpretation method combining overall and local features of cervical cells. From the experimental results, the classification accuracy was slightly higher for each outlet and the final integration than for the original method. It was demonstrated that the fusion of contextual building blocks and self-distillation methods can significantly improve the classification accuracy of shallow classifiers. From the integration strategy, the accuracy of the original method did not significantly improve compared with that of the sub-classifiers, while the method in this paper further improved the optimal sub-model accuracy by 0.47% through the integration strategy. It was found that the fused global context module ensured that the sub-classifiers focused more attention on the fine-grained features of different regions and this differentiation guaranteed the effectiveness of the integration.

### 4.4. Subjective Effect Analysis

To independently verify the effectiveness of the multi-export self-distillation method fusing contextual information proposed in this paper, experiments were conducted on the cervical cell and public datasets using different teacher–student networks and the results were compared with the use of mainstream knowledge distillation algorithms. These results are shown in Table 5 and Table 6. Compared with the traditional teacher–student network structure, the distillation method did not need the teacher network and only constructed classifiers at different stages of the student network to achieve self-distillation. The number of student network parameters increased from 11.22 M to 12.33 M and, despite this increase, the training cost of the teacher network was reduced, improving classification performance at the same time. From the results on the SIPaKMeD dataset, it can be seen that the model classification accuracy improved by 2.56 and 0.37% for the benchmark and knowledge distillation methods, respectively, compared with the proposed method. Thus, the performance of the ResNet18-based multi-export student network surpasses that of the teacher network.

On the CIFAR100 dataset, the proposed method improved 4.29 and 3.47% compared to the benchmark method, which also outperformed mainstream knowledge distillation. Improved performance was also achieved by fusing contextual information with multi-export self distillation without teacher network guidance. The experimental results demonstrate that the method significantly improved classification of both cervical cell and natural images.

## 5. Conclusions

In this paper, a transfer method, combining transfer learning and distillation, was proposed for cervical cell classification that transfers feature knowledge between the model and data so that the neural network can achieve improved model generalization using a limited number of samples. Furthermore, a new method for cervical cell classification with improved multi-export integration was proposed. A multi-export classification network was introduced to construct branching export network pairs using different depths to learn different fine-grained features. The global context module and the self-distillation algorithm were fused in each branch network to improve the performance of shallow classifiers. The final prediction was achieved using an integration strategy. From the experimental results, ResNet18, incorporating the methods of this paper, achieved 98.52% classification accuracy on the cervical cell dataset. These findings are of significant academic value and social significance for the intelligent classification of cervical cytology smear images.

In future work, we aim to extend the distillation method that combines knowledge distillation and self-distillation to more tasks. Combining self-distillation with other self-supervised distillation methods is a further possible research direction. In addition, the balance between performance improvement and memory usage needs to be considered.

## Figures and Tables

**Figure 1 biomimetics-07-00195-f001:**
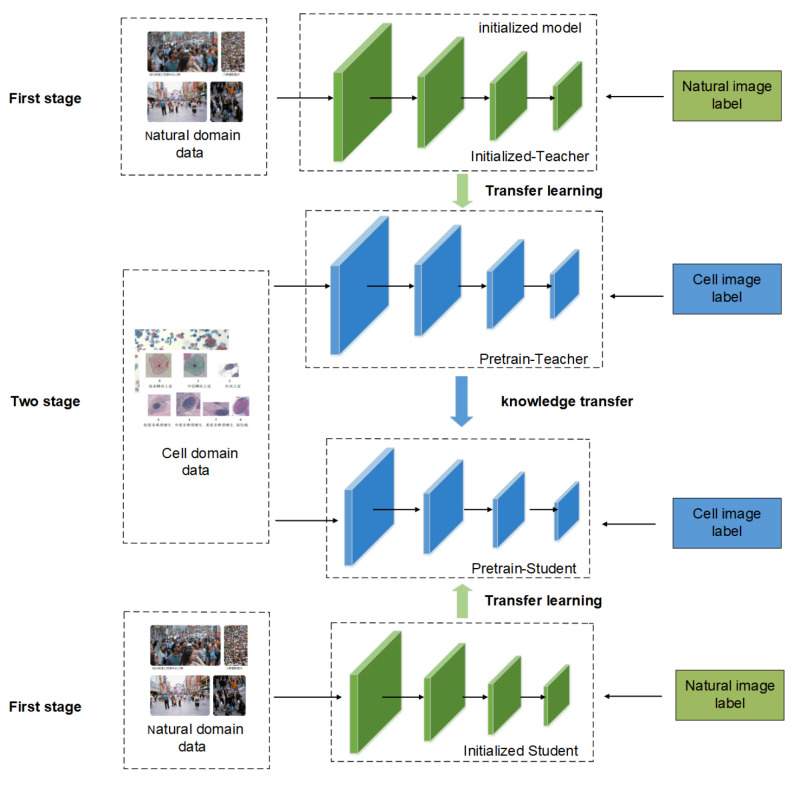
Schematic diagram of the transfer method framework and training process.

**Figure 2 biomimetics-07-00195-f002:**
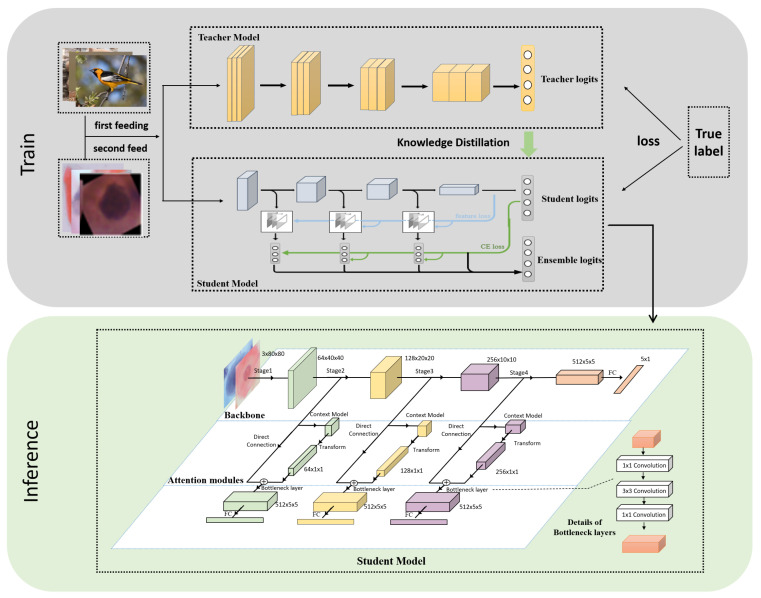
Overview of the proposed knowledge distillation-based method for cervical cell classification. The whole framework is divided into two parts: training and inference. The training part consists of: (i) pre-training the teacher and student networks first; (ii) fine-tuning using cervical cell images; (iii) the student network adding a bottleneck layer and a fully connected layer after each block to build a multi-exit network from shallow to deep, and (iv) in the inference stage, each classifier being combined in an ensemble to form a strong classifier.

**Figure 3 biomimetics-07-00195-f003:**
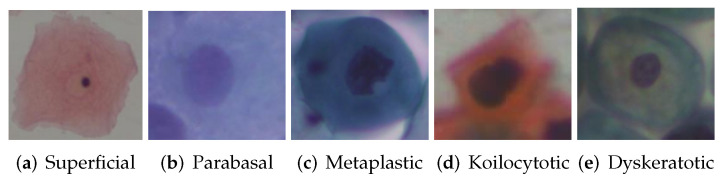
Examples of SIPaKMeD dataset image.

**Figure 4 biomimetics-07-00195-f004:**
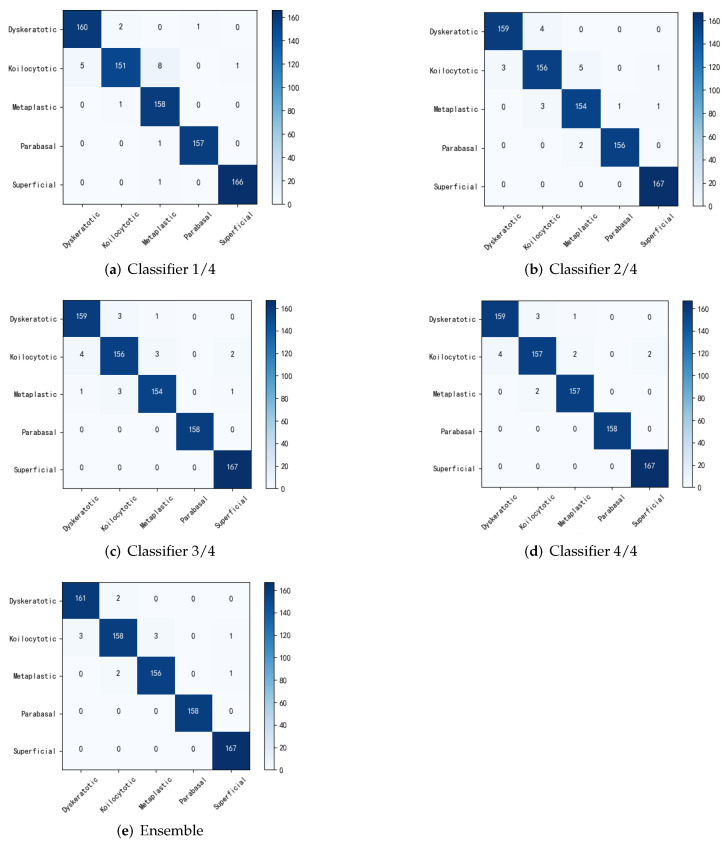
Confusion matrix for classification at different stages.

**Table 1 biomimetics-07-00195-t001:** SIPaKMeD dataset.

Dataset	Category	Cell Type	Quantity
SIPaKMeD	1	Superficial- Intermediate	831
	2	Parabasal	787
	3	Metaplastic	793
	4	Koilocytotic	825
	5	Dyskeratotic	813

**Table 2 biomimetics-07-00195-t002:** The accuracy of the proposed method compared with other methods using SIPaKMeD.

Model	Accuracy	Sensitivity	Specificity	F-Measure
DeepPap	93.58 ± 0.16	97.40 ± 0.80	96.90 ± 1.40	97.60 ± 0.50
MLP	88.54 ± 5.60	-	-	-
Deep convolutional+SVM	93.35 ± 0.62	-	-	-
Deep fully-connected+SVM	94.44 ± 1.21	-	-	-
CNN(VGG19)	95.35 ± 0.42	-	-	-
Mor-27	88.47 ± 0.92	95.90 ± 1.20	90.90 ± 1.10	95.00 ± 0.60
ResNet-101	94.86 ± 0.74	99.10 ± 0.70	97.70 ± 0.80	98.70 ± 0.20
DenseNet-121	96.79 ± 0.42	99.00 ± 0.50	98.90 ± 0.50	99.20 ± 0.10
GNC	97.37 ± 0.57	99.60 ± 0.10	99.50 ± 0.40	99.60 ± 0.20
ResNet50	97.63 ± 0.39	97.62 ± 1.25	99.50 ± 0.38	98.43 ± 0.40
Inception V3	97.72 ± 0.65	97.62 ± 1.15	98.83 ± 0.99	98.24 ± 0.64
Compact VGG	97.80 ± 0.50	97.80 ± 0.50	99.17 ± 0.57	98.28 ± 0.83
Ours Improved-ResNet18	98.52 ± 0.31	98.53 ± 0.35	98.68 ± 0.46	98.59 ± 0.23

**Table 3 biomimetics-07-00195-t003:** Experimental results of ablation with different strategies on SIPaKMeD.

Method	Accuracy
ResNeSt50 + TL	98.15%
ResNet18	95.31%
ResNet18 + TL	96.54%
ResNet18 + TL+KD	97.66%
ResNet18 + TL+KD+SD	98.52%

**Table 4 biomimetics-07-00195-t004:** Classification accuracy by different exit and ensemble.

Method	Exit-1	Exit-2	Exit-3	Exit-4	Ensemble
Baseline	92.61%	95.44%	94.82%	95.31%	95.93%
SCAN [31]	95.13%	95.43%	97.07%	96.32%	97.21%
MSD [36]	96.76%	96.28%	95.66%	96.23%	96.47%
Ours	95.65%	96.91%	97.16%	97.68%	98.15%

**Table 5 biomimetics-07-00195-t005:** Comparison of accuracy between the proposed and existing knowledge-distillation methods on SIPaKMeD.

Method	Accuracy
ResNeXt101	97.04%
ResNet18	95.56%
KD [23]	96.54%
FIT [24]	97.78%
AT [37]	97.16%
SD [38]	96.79%
VID [39]	96.42%
Ours	98.15%

**Table 6 biomimetics-07-00195-t006:** Comparison of accuracy between the proposed and existing knowledge-distillation methods on CIFAR100.

Teacher	ResNet152	ResNet152
Student	ResNet50	ResNet18
Baseline	80.91%	80.91%
	77.98%	77.09%
KD [23]	79.69%	79.86%
FIT [24]	80.51%	79.24%
AT [37]	80.41%	80.19%
VID [39]	79.24%	79.67%
RKD [40]	80.22%	79.60%
PKT [41]	80.57%	79.44%
AB [42]	81.21%	79.50%
CRD [43]	80.53%	79.81%
SSKD [44]	80.29%	80.36%
Ours	82.27%	80.56%

## Data Availability

The SIPaKMeD dataset is a publicly available dataset that can be found at https://www.cse.uoi.gr, accessed on 5 November 2022.

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
