# Peer review of "Cervical Cell Image Classification-Based Knowledge Distillation"

_biomimetics, 2022, doi:10.3390/biomimetics7040195_

Round 1

Reviewer 1 Report

1. In the Abstract section,it is suggested to add some other evaluation metrics, such as precision, sensitivity, recall, specificity, F1-score and G-mean. This is convenient for readers to determine whether the proposed method in this paper has advantages over other methods at the first time.

2. In the Introduction section, the authors just gave an example of the research results of the research content in this paper on deep learning. It is difficult to understand the disadvantages of traditional machine learning compared with deep learning in this direction only by description. It is recommended to add detailed comparison of evaluation metrics with numerical values.

3. In the section 2.2, Knowledge distillation strategies include hard distillation and soft distillation. The authors chose a method closer to soft distillation, Will the more direct hard distillation bring better results to the experiment?

4. In the section 3.2, the introduction of the self-distillation is slightly simple. It is suggested to add more contents to explain the principle and structure of the self-distillation.

5. In the section 4.1, generally speaking, the distillation temperature value in the knowledge distillation strategy is usually used to change the probability distribution. The temperature coefficient pulls in the probability between various categories, and the authors mentioned that T=3 was selected based on experience. However, this may not mean that 3 is the most suitable temperature coefficient for this experiment. It is suggested to add temperature τ comparison experiments to select the best τ.

6. Section 4.2 of the article refers to the comparison with the parameter quantity of the teacher model. However, the knowledge distillation training strategy needs to call the weights of the teacher model. What is the highest accuracy value if they only use the teacher model rather than the ResNet18 to train this task. Will the performance be better than that of the backbone ResNet18 without the help of the teacher model?

7. In the Section 4.2, the authors mentioned that the network parameters of this experiment are about 11M-12M. It is suggested that the authors add comparison and contrast with other methods in the number of parameters to highlight the advantages of the proposed method in parameter redundancy.

8. In the whole manuscript, the authors only used the accuracy rate as the evaluation metric, and it is recommended to add other evaluation metrics.

9. It is suggested that the authors write some ideas on how to further improve their results in the future work.

Author Response

Reviewer#1, Concern # 1:  In the Abstract section,it is suggested to add some other evaluation metrics, such as precision, sensitivity, recall, specificity, F1-score and G-mean. This is convenient for readers to determine whether the proposed method in this paper has advantages over other methods at the first time

Author response:  Thanks for your suggestion, we have added it in the revised version. According to the confusion matrix, we additionally calculated the important evaluation metrics of sensitivity, specificity, and F1-score, and compared them with the work of others. The results can be seen in Table 2.

Reviewer#1, Concern # 2:  In the Introduction section, the authors just gave an example of the research results of the research content in this paper on deep learning. It is difficult to understand the disadvantages of traditional machine learning compared with deep learning in this direction only by description. It is recommended to add detailed comparison of evaluation metrics with numerical values.

Author response:  Thanks for your suggestion, the method in this paper is based on deep learning. In the introduction, we describe traditional machine learning methods on the one hand and deep learning methods on the other. Among them, we compare several typical traditional machine learning methods. For example, the one-stage approach of [2] requires manual feature selection for SVM classification. And the two-stage method Otsu is used for threshold segmentation, followed by K-means clustering. On these foundations, according to your suggestion, we continue to investigate and supplement some related machine learning methods, such as Mor27 in Table 2. For evaluation metrics, there is a lack of comparability due to the fact that some methods are experimented on different datasets. Therefore, we only compare several machine learning works on the dataset in this paper, and compare them in Table 2 of the experimental results.

Reviewer#1, Concern # 3:  In the section 2.2, Knowledge distillation strategies include hard distillation and soft distillation. The authors chose a method closer to soft distillation, Will the more direct hard distillation bring better results to the experiment?

Author response:  Thank you for your question, at which point we need further explanation. Hard distillation and soft distillation refer to the smoothness of the output label of the teacher network, which is determined by the hyperparameter T. As T tends to infinity, it can be assumed that the predicted scores for each class are the same, which is not instructive. When T is 1, it is a standard softmax function output, which is hard distillation. When Professor Hinton first proposed knowledge distillation, he introduced the temperature T, and proved that using soft target can obtain more information to improve the effectiveness of distillation. Therefore, we did not try not to use the soft target as this violates the original knowledge distillation, but in future work, we further discuss the effect of T on distillation.

[1] HINTON G, VINYALS O, DEAN J. Distilling the Knowledge in a Neural Network[J], 2015.

Reviewer#1, Concern # 4:  In the section 3.2, the introduction of the self-distillation is slightly simple. It is suggested to add more contents to explain the principle and structure of the self-distillation.

Author response:  Thanks for the suggestion, we are indeed missing the discourse on self-distillation. Therefore, we supplement the details related to self-distillation in the section 3.2, and also supplement some work on self-distillation in the related work. This makes our paper look more complete.

Reviewer#1, Concern # 5:  In the section 4.1, generally speaking, the distillation temperature value in the knowledge distillation strategy is usually used to change the probability distribution. The temperature coefficient pulls in the probability between various categories, and the authors mentioned that T=3 was selected based on experience. However, this may not mean that 3 is the most suitable temperature coefficient for this experiment. It is suggested to add temperature τ comparison experiments to select the best τ.

Author response: As explained in question 3 above, T acts as a degree coefficient similar to label smoothing. Not in extreme cases, it has less impact on experiments. Therefore, in many knowledge distillation methods, the value of T is not deliberately adjusted to obtain a good distillation effect. Likewise, in our work, we keep the parameter settings for T in the conventional distillation method without deliberately picking an optimal one. We mainly want to show the effectiveness of the method. In future work, we will consider it. Thanks for your advice.

Reviewer#1, Concern # 6:  Section 4.2 of the article refers to the comparison with the parameter quantity of the teacher model. However, the knowledge distillation training strategy needs to call the weights of the teacher model. What is the highest accuracy value if they only use the teacher model rather than the ResNet18 to train this task. Will the performance be better than that of the backbone ResNet18 without the help of the teacher model?

Author response:  Thank you for your suggestion, indeed, during the training process, we need to use the weights of the teacher network to train the student network (ResNet18), which increases the training cost. However, predictive inference is performed with the student network alone. Usually, when the model is deployed, the reasoning process of the algorithm will be paid more attention, so we also compare the parameters of ResNet18 with other methods.

Regarding the performance of using only the teacher network or the student network, we have demonstrated the ablation experiments in the paper. As shown in Table 3, the accuracy rate of using only the teacher network (ResNeSt50) is 98.15%, and the accuracy rate of the student network(ResNet18) without the help of the teacher network is 95.31%, obviously, the performance of the teacher network is better than that of the student network, but through the algorithm in this paper, the performance of the student network is improved by 3.21%, and it exceeds the teacher network at the same time.

Reviewer#1, Concern # 7:  In the Section 4.2, the authors mentioned that the network parameters of this experiment are about 11M-12M. It is suggested that the authors add comparison and contrast with other methods in the number of parameters to highlight the advantages of the proposed method in parameter redundancy.

Author response: Thanks for the reminder, it's a good suggestion for us to improve the paper. However, since some of the work is not open source, we cannot supplement the parameters of other models. If we try to estimate, there will be some errors in the calculation of parameters, which may be imprecise.

Reviewer#1, Concern # 8:  In the whole manuscript, the authors only used the accuracy rate as the evaluation metric, and it is recommended to add other evaluation metrics.

Author response:  Thank you for your correction, as answered in Question 1, we have added multiple evaluation metrics to verify the effectiveness of our method, and the results are shown in Table 2. In addition we also added additional traditional methods such as Mor27. From the latest results, it can be seen that the accuracy of our small network (ResNet18) is still the best, and several other indicators are second only to the best results, which is also the direction of the follow-up efforts of this paper.

Reviewer#1, Concern # 9: It is suggested that the authors write some ideas on how to further improve their results in the future work.

Author response:  This is indeed our mistake, we have added it at the end of the article, thank you for your suggestion.

Reviewer 2 Report

This paper aims at developing an end-to-end model to implement cervical cell classification based on deep learning, which has good clinical application value. The results look promising. However, the paper may have the following problems, which can be improved.

Q1:

In the Introduction part. Highlight the innovation points of the paper. Give what harm parameter redundancy will bring to the deep learning model?

Q2:

Some advances of deep learning in biomedical image analysis are not mentioned, as follows:

[1] Source-free unsupervised domain adaptation for cross-modality abdominal multi-organ segmentation. Knowledge-Based Systems, 2022, 109155. DOI: 10.1016/j.knosys.2022.109155

[2] Unsupervised domain adaptation for cross-modality liver segmentation via joint adversarial learning and self-learning. Applied Soft Computing, 2022, 121: 108729. DOI: 10.1016/j.asoc.2022.108729

[3] Improvement of cerebral microbleeds detection based on discriminative feature learning. Fundamenta Informaticae, 2019, 168(2-4): 231-248. DOI: 10.3233/FI-2019-1830

[4] Brain Age Prediction of Children Using Routine Brain MR Images via Deep Learning. Frontiers in Neurology, 2020. DOI: 10.3389/fneur.2020.584682.

Q3:

In Figure 1, I only see transfer learning, but I can't see where the distillation of knowledge is reflected

Q4:

“moreover, individual networks cannot usually learn all view features due to their limited "capacity". Therefore, based on transfer learning, we introduced a knowledge distillation algorithm, i.e., we constructed a teacher–student network to transfer category information learned in the teacher to the student network and improved its performance.” I am not sure that knowledge distillation can provide more view features, so students' network performance is improved? Pls give the corresponding Ref.

Q5:

Figure 2 fails to explain the method in this paper.

It is very strange to input natural images and cervical cell images to the two networks (teacher and student) at the same time during the training process, and that's what this figure means.

In the knowledge distillation method, the teacher network only needs to provide soft tags to the student network, but the figure shows that the weight of the entire teacher network is indeed shared with the student network.

Whose label is the “Ture label”? Natural image or medical cell image. If there is no "true label" of a natural image, what is the purpose of inputting natural images?

The figure must be completely reconstructed.

In addition, the figure used to describe the full-text method cannot only have the name of the figure.

Q6:

This paper divides the dataset according to the ratio of 3:1:1. Are other methods in Table 2 also divided according to this ratio.

Q7:

From the prediction results in Table 2 and Table 6, the performance of the student network does not exceed that of the teacher network (even if the number is a little bit larger, it may only appear randomly, because the authors did not test the significant difference between the two prediction results, such as P value). My understanding is that the purpose of knowledge distillation is to reduce model parameters without reducing performance as much as possible. It is not expected that the performance of student network will exceed that of teacher network. I think the author's wording may need to take note of this.

Q8:

In Table 3, Pure transfer learning should be ResNet50+TL, which should be added to the table.

Q9:

The paper has some grammatical errors and some technical terminology errors (such as migration learning).

Author Response

Reviewer#2, Concern # 1:  In the Introduction part. Highlight the innovation points of the paper. Give what harm parameter redundancy will bring to the deep learning model?

Author response:  Thanks for your suggestion, we have added relevant content in the highlighted part of the introduction.

Reviewer#2, Concern # 2:  Some advances of deep learning in biomedical image analysis are not mentioned, as follows:

[1] Source-free unsupervised domain adaptation for cross-modality abdominal multi-organ segmentation. Knowledge-Based Systems, 2022, 109155. DOI:10.1016/j. knosys.2022.109155

[2] Unsupervised domain adaptation for cross-modality liver segmentation via joint adversarial learning and self-learning. Applied Soft Computing, 2022, 121: 108729. DOI:10.1016/j.asoc.2022.108729

[3] Improvement of cerebral microbleeds detection based on discriminative feature learning. Fundamenta Informaticae, 2019,168(2-4): 231-248. DOI: 10.3233/FI-2019-1830

[4] Brain Age Prediction of Children Using Routine Brain MRI mages via Deep Learning. Frontiers in Neurology, 2020. DOI:10.3389/fneur.2020.584682.

Author response: We have cited and inspired the above papers, which are really worth studying.

Reviewer#2, Concern # 3: In Figure 1, I only see transfer learning, but I can't see where the distillation of knowledge is reflected.

Author response: Thank you so much for asking this question, it's the question we've formulated. We have re-adjusted Figure 1 to make transfer learning and knowledge distillation more intuitive, hoping to clear your doubts.

Reviewer#2, Concern # 4:  moreover, individual networks cannot usually learn all view features due to their limited "capacity". Therefore, based on transfer learning, we introduced a knowledge distillation algorithm, i.e., we constructed a teacher–student network to transfer category information learned in the teacher to the student network and improved its performance.” I am not sure that knowledge distillation can provide more view features, so students' network performance is improved? Pls give the corresponding Ref.

Author response:  Thanks, that's a really good question. There is still a lot of work so far on how to understand knowledge distillation, which also means that the effectiveness of knowledge distillation has been a controversial topic, and we also try to explain it. Prior to this, we read a piece of work from Microsoft Research and Kainegie Mellon University, who argued that knowledge distillation is forcing the student network to learn the different view characteristics of the teacher network, and to argue. We have reached some consensus, so the explanation of the validity of the method in this paper draws on some points of Allen-Zhu Z et al. In the revised version, we add a citation note.

[1] Allen-Zhu Z, Li Y. Towards understanding ensemble, knowledge distillation and self-distillation in deep learning[J]. arXiv preprint arXiv:2012.09816, 2020.

Reviewer#2, Concern # 5:  Figure 2 fails to explain the method in this paper. It is very strange to input natural images and cervical cell images to the two networks (teacher and student) at the same time during the training process, and that's what this figure means. In the knowledge distillation method, the teacher network only needs to provide soft tags to the student network, but the figure shows that the weight of the entire teacher network is indeed shared with the student network. Whose label is the “Ture label”? Natural image or medical cell image. If there is no "true label" of a natural image, what is the purpose of inputting natural images? The figure must be completely reconstructed. In addition, the figure used to describe the full-text method cannot only have the name of the figure.

Author response:  Thank you very much for your question, which is confusing to you due to our unclear presentation. We have restructured all diagrams in this paper describing the method, including Figures 1 and 2, and added the necessary descriptions.

Also, explain your question here. Our method can be better understood through the modified Figure 1, in order to implement transfer learning. Natural image data and cervical cell image data enter the network in two stages. The first stage is to initialize the model pre-training with natural image pairs, which includes both the student network and the teacher network. Apparently the true labels at this time come from natural images. The second stage is to use the cervical cell image data to fine-tune the pre-trained network to achieve the purpose of model weight reuse. This stage is to use the ground truth labels of the cell data. Then there is the distillation stage. The first is to freeze the teacher network (not participating in training), and only use the soft tags output by the teacher network and the soft tags of the student network to calculate the loss function, so as to achieve the purpose of supervising the student network by the teacher network, thereby improving students network performance. However, in our original Figure 2, we tried to use the blue arrow sign to represent the above distillation process, which led to the ambiguity of model weight sharing. We fully recognized the inadequacy of the image representation, and also restructured it, corrected some lines in Figure 2, and added annotations, hoping to make it easier to understand. Thanks again for your attentiveness.

Reviewer#2, Concern # 6:  This paper divides the dataset according to the ratio of 3:1:1. Are other methods in Table 2 also divided according to this ratio.

Author response:  Several other methods do not set up a test set, they divide the data into five parts, choose one part as the validation set in turn, and the rest as training samples. We believe that the dataset contains 4096 images, which is not a small number in medical images, so the data division rules of conventional natural images are adopted. In comparison, we achieve better test accuracy with fewer training samples. We have to admit that we did not achieve the consistency of data partitioning, but it is also very difficult due to the randomness of partitioning. Thanks for your question, we will take note of this in future work.

Reviewer#2, Concern # 7:  From the prediction results in Table 2 and Table 6, the performance of the student network does not exceed that of the teacher network (even if the number is a little bit larger, it may only appear randomly, because the authors did not test the significant difference between the two prediction results, such as P value). My understanding is that the purpose of knowledge distillation is to reduce model parameters without reducing performance as much as possible. It is not expected that the performance of student network will exceed that of teacher network. I think the author's wording may need to take note of this.

Author response:  Indeed, compared to the teacher network, the student network is not good enough, but comparing the performance of the independently trained student network, we have improved the cervical cell prediction accuracy from 95.31% to 98.52%. On cifar100, the student network was changed from 77.98% and 77.09% to 82.27% and 80.56%, respectively, which has a positive effect.

Regarding the discussion of knowledge distillation, thank you for your suggestion, and we have made corresponding changes in the paper. As you said, the original idea of knowledge distillation is for model compression, rather than focusing on improving performance. However, in recent years, there has been some work on self-distillation, which uses distillation to improve model performance without the need for a teacher network. In this paper, it is the combination of these two approaches. The specific approach is to change the student network to a multi-exit network on the traditional distillation architecture. While the teacher network provides supervision information, it optimizes itself through self-distillation. It not only improves the performance of the model, but also achieves a certain compression rate.

Reviewer#2, Concern # 8:  In Table 3, Pure transfer learning should be ResNet50+TL, which should be added to the table.

Author response: Thank you for your careful review. In Table 3, we mainly want to show the experimental results of student network ablation, and analyze the various strategies applied on the student network one by one. Because the student network is a key component of the entire structure. Therefore, the teacher network is not considered too much. The ResNeSt50 in the table is the final model used to guide the training of the student network, which is the result of using transfer learning. This network is a variant of ResNet50 that combines the Split-Attention module. For further specification, we make modifications in Table 3. Changed the original ResNeSt50 to ResNeSt50+TL, which is indeed the case.

Reviewer#2, Concern # 9:  The paper has some grammatical errors and some technical terminology errors (such as migration learning).

Author response:  This was indeed a huge mistake for the overall quality of our article. We have corrected it and we have changed the migration learning to transfer learning. Thank you very much for pointing it out.

Round 2

Reviewer 1 Report

For the issues raised in the review proposal, the authors have revised and explained the reasons for each issue. I recommend the editorial department to accept it.

Reviewer 2 Report

All suggestions have been addressed